# Zero-Shot Robustness of Vision Language Models Via Confidence-Aware Weighting

**Nikoo Naghavian**
School of ECE, College of Engineering
University of Tehran
nikoo.naghavian@ut.ac.ir

**Mostafa Tavassolipour**
School of ECE, College of Engineering
University of Tehran
tavassolipour@ut.ac.ir

## Abstract

Vision-language models like CLIP demonstrate impressive zero-shot generalization but remain highly vulnerable to adversarial attacks. In this work, we propose Confidence-Aware Weighting (CAW) to enhance zero-shot robustness in vision-language models. CAW consists of two components: (1) a Confidence-Aware loss that prioritizes uncertain adversarial examples by scaling the KL divergence between clean and adversarial predictions, and (2) a feature alignment regularization that preserves semantic consistency by minimizing the distance between frozen and fine-tuned image encoder features on adversarial inputs. These components work jointly to improve both clean and robust accuracy without sacrificing generalization. Extensive experiments on TinyImageNet and 14 additional datasets show that CAW outperforms recent methods such as PMG-AFT and TGA-ZSR under strong attacks like AutoAttack, while using less memory.

## 1 Introduction

Traditional deep learning approaches rely on pre-training followed by fine-tuning with labeled data for each downstream task. The emergence of GPT-3 [1] in the natural language processing field has popularized models with zero-shot capability, where models trained on diverse internet-scale data can be applied to a wide range of tasks and unseen domains. In the multimodal setting, CLIP [2] employs a contrastive loss [3, 4] to align matching image–text pairs in a shared embedding space while separating mismatched pairs. This enables the model to acquire broad vision–language knowledge and achieve strong performance across various tasks, including image classification [5, 6], semantic segmentation [7], object detection [8, 9], image–text retrieval [10], and visual question answering [11]. Although CLIP demonstrates strong generalization ability, it remains vulnerable to small, imperceptible perturbations that leave the image visually unchanged to humans but cause significant shifts in predictions [12]. Adversarial training [13] is among the most effective approaches for improving robustness against strong attacks, typically training from scratch with both adversarial and clean examples. However, when applied to large-scale models like CLIP, adversarial training must be adapted to prevent overfitting and the forgetting of pre-trained knowledge, while still enhancing robustness [14, 15, 16].

The TeCoA [16] method was the first to study the zero-shot robustness of large-scale vision-language models. It showed the importance of using text supervision with a contrastive adversarial loss while applying different adaptation approaches [17]. Later, the PMG-AFT [18] method added new terms to the previous loss function to enhance robustness while causing a smaller decrease in performance on clean data. More recently, TGA-ZSR [19] introduced a method that improves both robustness and clean accuracy, along with the interpretability of attacks. This approach used text supervision with semantic information instead of relying on the model's output probabilities. Despite their

Accepted to the NeurIPS 2025 Workshop on Reliable ML from Unreliable Data.

effectiveness, these methods either need high memory usage, or still struggle to maintain robust accuracy under strong attacks.

We propose a novel adversarial fine-tuning loss named Confidence-Aware Weighting (CAW) that improves the robustness of a pre-trained CLIP model while preserving clean accuracy and reducing memory usage. This method introduces two key components designed to improve robustness and maintain generalization. The first is a Confidence-Aware term, which weights the KL divergence between clean and adversarial prediction distributions of the fine-tuned and frozen pre-trained CLIP models, ensuring that training focuses more on hard adversarial examples. The second is a regularization term, which matches adversarial image features from the fine-tuned image encoder with those from the frozen pre-trained encoder, helping retain semantic knowledge from the pre-trained model and reducing overfitting. Experiments on TinyImageNet and 14 zero-shot datasets (see Appendix B for details) demonstrate state-of-the-art performance under AutoAttack, surpassing both PMG-AFT and TGA-ZSR in robust accuracy. Under PGD-100 and CW, the proposed method outperforms PMG-AFT in both robust and clean accuracy, while maintaining lower memory usage than both baselines.

The key contributions of this work are:

- Propose CAW to improve zero-shot robustness by emphasizing challenging samples.
- Achieves higher robust accuracy than PMG-AFT and TGA-ZSR under AutoAttack.
- Improves clean and robust accuracy over PMG-AFT under PGD-100 and CW.
- Requires less memory than PMG-AFT and TGA-ZSR.

## 2 Methodology

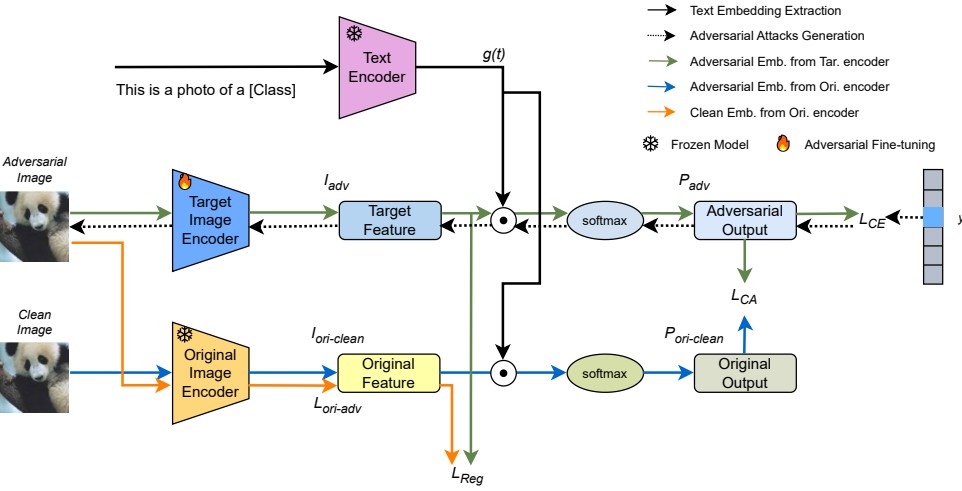

Figure 1: Overview of Confidence-Aware Weighting (CAW) method. $\odot$ means matrix inner product.

### 2.1 Preliminaries and Problem Setup

In this work, we employ CLIP [2] to enhance zero-shot robustness in classification tasks. CLIP has two encoders that learn a joint visual–text feature space. At inference, the predicted label is the one that text embedding has the highest cosine similarity with the image embedding. Following prior works [19, 18], we fine-tune the model using the cross-entropy loss:

$$L_{CE}(x, t, y) = -\mathbb{E}_{i,j} \left[ y_{ij} \log \frac{\exp(\cos(f(x)_i, g(t)_j)/\tau)}{\sum_k \exp(\cos(f(x)_i, g(t)_k)/\tau)} \right] \tag{1}$$

where $f(x)$ and $g(t)$ denote the image and text embeddings, $\tau$ is the temperature parameter, and $\cos$ is the cosine similarity. The label $y_{ij}$ is set to 1 for positive image-text pairs and 0 otherwise.

### 2.1.1 Adversarial Attacks

Deep learning models are typically trained and evaluated on clean data; however, small, imperceptible perturbations can cause significant prediction errors. Such perturbations can be generated using attack methods including PGD [20], AutoAttack [21], and CW [22]. PGD is an iterative attack that applies noise over multiple steps, producing stronger adversarial examples $x_a$ than single-step methods such as FGSM [12]. It seeks perturbations that maximize the loss while keeping the perturbed input within a specified neighborhood of the clean example:

$$x_{a+1} = \Pi_{x+\mathcal{S}} \left( x_a + \varepsilon \cdot \text{sign} \left( \nabla_{x_a} L(x_a, t, y) \right) \right) \tag{2}$$

where $L$ denotes the loss function, $x$ is the clean input, $\varepsilon$ is the perturbation bound under the $p$-norm, and $\nabla_x L$ is the gradient direction that increases the loss. The set $\mathcal{S}$ represents the allowed changes that the adversary can make to the input.

### 2.1.2 Adversarial Training

By optimizing over adversarially perturbed inputs, adversarial fine-tuning enables models to learn more robust features through a min-max objective. In some cases, including our method, the objective function used to craft adversarial examples differs from the one used to optimize the model parameters. Specifically, in the inner loop (Equation 3), adversarial examples $x_a$ are generated by maximizing the loss $L$ (i.e., $\mathcal{L}_{\text{CE}}$, as defined in Equation 1 of our method), which is optimized using the PGD update rule (Equation 2). In the outer loop (Equation 4), the model parameters $\theta$ are updated by minimizing a separate loss function $\mathcal{J}$ (i.e., $\mathcal{L}_{\text{total}}$, as defined in Equation 10 of our method).

$$x_a = \arg\max_{x} \; L(x, t, y) \tag{3}$$

$$\theta^\star = \arg\min_{\theta} \; \mathcal{J}_\theta(x_a, t, y) \tag{4}$$

## 2.2 Method

Building on previous studies [16, 18, 19], we aim to preserve the generalizable and robust features learned by the pre-trained CLIP model during fine-tuning with a new loss function. As illustrated in Figure 1, we use both the original and target image encoders to retain prior knowledge while improving robustness. Although the TeCoA method [16] introduces a contrastive loss using adversarial examples with text supervision, it remains insufficient for jointly improving clean and robust accuracy. To address this limitation, we propose two additional loss terms that enhance robustness while maintaining generalization to unseen tasks.

**Confidence-Aware Term** We propose Confidence-Aware loss that focuses on challenging samples by emphasizing hard adversarial examples, i.e., those where the model is less confident in the correct class, while down-weighting easier ones. In contrast to prior methods that treat all samples equally in the loss function, our approach explicitly targets the inherent weaknesses in adversarial training by assigning more weight to samples that are more easily fooled by adversaries. This idea is inspired by the ARoW method [23], which prioritizes vulnerable samples to enhance adversarial robustness. However, our formulation significantly differs in both design and scope, as it is tailored to the unique challenges of vision-language models and zero-shot generalization. Specifically, we define a KL-based alignment between the frozen CLIP model's predictions on clean images, $P^{\text{clean}}$, and the fine-tuned model's predictions on adversarial images, $P^{\text{adv}}$. This alignment allows the model to retain semantic knowledge from pre-training while learning to handle difficult adversarial examples. Unlike ARoW, which uses the reverse KL divergence ($\text{KL}(P^{\text{clean}} \| P^{\text{adv}})$), we place the adversarial distribution as the first argument, i.e., $\text{KL}(P^{\text{adv}} \| P^{\text{clean}})$, which showed better results in our experiments. The distributions $P^{\text{adv}}$ and $P^{\text{clean}}$ are defined as:

$$P^{\text{adv}} = \text{softmax}(f(x_{\text{adv}})_{\text{tar}} \cdot g(t)^\top), \tag{5}$$

$$P^{\text{clean}} = \text{softmax}(f(x_{\text{clean}})_{\text{ori}} \cdot g(t)^\top), \tag{6}$$

where $f(\cdot)$ and $g(\cdot)$ denote the image and text encoder embeddings, and the dot operator represents the matrix inner product between these embeddings. The subscripts `tar` and `ori` refer to features from the fine-tuned and frozen image encoders. The element $P^{\text{adv}}_{i,y_i}$ denotes the predicted probability for the true label $y_i$ under the adversarial input $x^{\text{adv}}_i$, as defined in Equation 7:

$$P^{\text{adv}}_{i,y_i} = \left[ \text{softmax} \left( f(x^{\text{adv}}_i) \cdot g(t)^\top \right) \right]_{y_i}. \tag{7}$$

Table 1: Zero-shot robust accuracy under AutoAttack with $\epsilon = 1/255$ on 15 datasets. We highlight the optimal accuracy in bold and underline the second-best result.

| Methods | Tiny-ImageNet | CIFAR-10 | CIFAR-100 | STL-10 | SUN-397 | Food101 | OxfordPets | Flowers102 | DTD | EuroSAT | FGVCAircraft | Caltech-101 | Caltech-256 | StanfordCars | PCAM | Average |
|---|---|---|---|---|---|---|---|---|---|---|---|---|---|---|---|---|
| CLIP | 0.02 | 0.01 | 0.08 | 0.03 | 0.04 | 0.01 | 0.00 | 0.03 | 0.16 | 0.12 | 0.06 | 0.43 | 0.10 | 0.11 | 0.22 | 0.09 |
| FT-Clean | 0.08 | 0.03 | 0.01 | 0.91 | 0.09 | 0.04 | 0.06 | 0.48 | 0.02 | 0.03 | 1.38 | 0.66 | 0.03 | 0.03 | | 0.26 |
| FT-Adv | 50.48 | 37.55 | 20.39 | 69.14 | 16.25 | 11.23 | 33.91 | 18.54 | **19.95** | 11.59 | 1.65 | 49.90 | 39.24 | 7.57 | **48.84** | 29.08 |
| TeCoA | 35.03 | 28.18 | 16.09 | 66.08 | 17.41 | 13.05 | 34.81 | 20.80 | 15.37 | 11.40 | 1.32 | 54.54 | 40.15 | 7.15 | 47.12 | 27.23 |
| FARE | 28.59 | 23.37 | 13.58 | 60.70 | 9.72 | 13.88 | 27.72 | 15.48 | 9.15 | 0.25 | 0.87 | 47.45 | 36.68 | 6.77 | 10.23 | 20.30 |
| PMG-AFT | 44.26 | 44.12 | 23.66 | 73.90 | 19.63 | 17.25 | 39.25 | 20.87 | 13.72 | **11.99** | 1.68 | 60.57 | 44.25 | 9.59 | 48.53 | 31.55 |
| TGA-ZSR | 49.45 | 40.53 | 22.38 | 72.06 | **20.36** | 15.58 | 40.31 | 21.43 | 17.13 | 11.19 | **2.64** | 57.16 | 45.68 | 10.47 | 48.03 | 31.63 |
| CAW | **50.52** | **47.35** | **26.35** | **74.27** | 19.64 | **20.50** | **41.89** | **21.61** | 16.80 | 11.11 | 2.52 | 62.79 | **47.27** | **12.23** | 47.81 | **33.51** |

To incorporate this into training, we minimize the KL divergence between $P^{\mathrm{adv}}$ and $P^{\mathrm{clean}}$, scaled by $1 - P^{\mathrm{adv}}_{i,y_i}$ to give greater importance to uncertain adversarial examples. This results in the Confidence-Aware loss:

$$L_{\mathrm{CA}} = \frac{1}{N} \sum_{i=1}^{N} \left[ \mathrm{KL}\left( P^{\mathrm{adv}}_i \,\|\, P^{\mathrm{clean}}_i \right) \left( 1 - P^{\mathrm{adv}}_{i,y_i} \right) \right]. \tag{8}$$

**Regularization Term**  We introduce a regularization loss that encourages consistency between the image encoder features of the frozen model, $f(\cdot)_{\mathrm{ori}}$, and the fine-tuned model, $f(\cdot)_{\mathrm{tar}}$, for adversarial inputs. This loss is computed before the text alignment stage, where the image features contain rich semantic information about the visual input. By aligning these features using the $\ell_2$ distance metric, the model retains the pre-trained CLIP knowledge and reduces the risk of overfitting during adversarial fine-tuning. The regularization loss is defined as:

$$L_{\mathrm{Reg}} = \frac{1}{N} \sum_{i=0}^{N} \left\| f(x_{\mathrm{adv}})_{\mathrm{tar}} - f(x_{\mathrm{adv}})_{\mathrm{ori}} \right\|_2. \tag{9}$$

The overall loss function is formulated as follows:

$$L_{\mathrm{total}} = L_{\mathrm{CE}} + \alpha \cdot L_{\mathrm{CA}} + \beta \cdot L_{\mathrm{Reg}}. \tag{10}$$

## 3 Experiments

**AutoAttack**  As shown in Table 1, our method outperforms all compared approaches under AutoAttack. On average, it achieves a 2% improvement in robust accuracy, demonstrating that the proposed training strategy learns transferable and more robust features resistant to this stronger attack. The model is trained with PGD-2 using a perturbation bound of $\epsilon = 1/255$ and evaluated on AutoAttack with the same perturbation bound. See Appendix A for related work, Appendix B for implementation details and datasets, Appendix C for additional experiments and ablation studies, and Appendix D for limitations and broader impact.

## 4 Conclusion

In this work, we demonstrate that emphasizing vulnerable samples during training improves the zero-shot robustness of CLIP. To this end, we introduce a CAW method that encourages the model to focus on hard adversarial examples, enabling the learning of more robust and transferable features. Experimental results show that our method outperforms prior approaches in both clean and robust accuracy across diverse domains under strong attacks, while requiring less memory, which is important for large-scale models. For future work, we aim to design a loss function that combines the idea of weighting challenging samples with attention mechanisms, which are essential components of large-scale models, to achieve better robustness against various attacks. Additionally, improving model interpretability by analyzing the features that contribute to robustness on difficult examples may provide deeper insights into building more resilient vision-language models.

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

# Appendix

## A  Related Work

**Adversarial robustness**  Deep neural networks have achieved remarkable performance on complex tasks, often producing highly confident predictions. However, small, imperceptible perturbations to the input can easily mislead them, resulting in incorrect outputs [20, 24, 25, 12, 26]. To address this vulnerability, various techniques have been proposed, including distillation [27], model compression [28], activation pruning [29], gradient regularization [30, 31] and adversarial training [20]. Adversarial training remains the most effective approach, augmenting adversarial examples alongside clean data during training to improve robustness while maintaining generalization [32]. Methods such as TRADES [15] balance clean accuracy and robustness by combining standard classification loss with a robustness regularization term. MART [33] highlights the importance of misclassified examples for improving robustness, while ARoW [23] focuses on the most vulnerable samples to enhance both generalization and robustness. HAT [34] mitigates over-robustness by introducing helper examples, arguing that pushing decision boundaries too far can harm clean accuracy.

**Zero-shot Adversarial Robustness for VLMs**  The introduction of the attention mechanism [35], combined with advances in GPUs and access to large-scale unlabeled internet data, enabled the development of language models like BERT [36], GPT-2 [37], and GPT-3 [1], marking a new era in deep learning. GPT-3's emergence brought zero-shot capabilities, allowing knowledge transfer to unseen domains and tasks. Following this trend, vision-language models (VLMs) such as BLIP [38], CLIP [2] and ALIGN [39] incorporate textual information with images to improve performance across diverse tasks rather than a single downstream application. Despite their generalization ability, VLMs remain vulnerable to imperceptible perturbations in the input, which can cause incorrect predictions[14]. Recent research has explored enhancing VLM robustness. TeCoA [16] introduced the use of text knowledge for model alignment with adversarial examples through contrastive loss. PMG-AFT [18] and TGA-ZSR [19] extended this approach by adding terms such as KL divergence or semantic alignment with text embeddings to improve both clean and robust accuracy. Another method [40] extracts normalized semantic feature embeddings (anchors) for each class label from a CLIP text encoder and uses them to guide the image encoder during adversarial training, enabling robustness transfer to unseen categories. FARE approach [41] aligns adversarial example features directly with the embeddings of a pre-trained CLIP model without requiring labels. Another related work [42] leverages not only the final adversarial examples from the PGD process but also intermediate samples along the adversarial trajectory for training. Our work focuses on challenging adversarial examples to guide the model in learning more robust and generalizable features.

## B  Datasets and Implementation

**Datasets**  To evaluate both clean and adversarial performance, we conduct extensive experiments on a diverse collection of image classification datasets. Our primary model, a pre-trained CLIP, is fine-tuned on the TinyImageNet [43] dataset. Evaluation is then performed not only on TinyImageNet but also on 14 additional datasets spanning five distinct domains. These include general object recognition benchmarks such as CIFAR-10 [44], CIFAR-100 [44], STL-10 [45], Caltech-101 [46], and Caltech-256 [47]; fine-grained classification datasets like OxfordPets [48], Flowers102 [49], FGVCAircraft [50], and StanfordCars [51]; scene recognition via SUN397 [52]; domain-specific datasets including Food101 [53], EuroSAT [54], and DTD [55]; and one medical imaging dataset, PCAM [56].

**Implementation Details**  For implementation, we use the ViT-B/32 architecture as the backbone for the CLIP model and fine-tune it on adversarial examples generated from the TinyImageNet dataset. Adversarial examples for both training and evaluation are produced using PGD attacks under the $\ell_\infty$ norm. Training updates all image encoder parameters using SGD with a learning rate of $1 \times 10^{-4}$, momentum of 0.9, weight decay of 0, and a batch size of 128. All experiments use PGD with 2 iterations and a perturbation bound of $1/255$. For evaluation, we employ PGD-100, AutoAttack, and CW, each with a step size equal to the perturbation bound. We set the hyperparameters $\alpha = 6$ and $\beta = 3$ to balance clean and robust accuracy. To ensure fair comparison, we adopt settings consistent with prior studies, which also used an RTX 3090 GPU.

Table 2: Zero-shot robust accuracy under PGD-100 with $\epsilon = 1/255$ on 15 datasets. All methods are fine-tuned on TinyImageNet using PGD-2.

| Methods | Tiny-ImageNet | CIFAR-10 | CIFAR-100 | STL-10 | SUN397 | Food101 | OxfordPets | Flowers102 | DTD | EuroSAT | FGVCAircraft | Caltech-101 | Caltech-256 | StanfordCars | PCAM | Average |
|---|---|---|---|---|---|---|---|---|---|---|---|---|---|---|---|---|
| CLIP | 0.88 | 2.42 | 0.26 | 26.11 | 1.00 | 6.60 | 3.84 | 1.19 | 2.02 | 0.05 | 0.00 | 19.88 | 12.60 | 0.20 | 0.11 | 5.14 |
| FT-Clean | 13.55 | 19.92 | 4.94 | 40.00 | 0.82 | 2.64 | 2.40 | 0.68 | 2.66 | 0.05 | 0.03 | 14.95 | 9.69 | 0.09 | 1.32 | 7.58 |
| FT-Adv. | 51.59 | 38.58 | 21.28 | 69.55 | 17.60 | 12.55 | 34.97 | 19.92 | 15.90 | 11.95 | 1.83 | 50.73 | 48.48 | 8.42 | 48.88 | 30.15 |
| TeCoA | 37.57 | 30.30 | 17.53 | 69.17 | 19.70 | 14.76 | 36.44 | 22.46 | 17.45 | 12.14 | 1.62 | 55.86 | 41.89 | 8.79 | 47.39 | 28.87 |
| FARE | 23.88 | 21.25 | 10.72 | 59.59 | 8.30 | 10.97 | 24.56 | 15.48 | 10.96 | 0.14 | 0.84 | 45.96 | 34.35 | 4.38 | 10.17 | 18.77 |
| PMG-AFT | 47.11 | 46.01 | 25.83 | 73.92 | 22.21 | 19.58 | 41.62 | 23.45 | 15.05 | 12.54 | 1.98 | 62.42 | 45.99 | 11.72 | 48.64 | 33.20 |
| CAW | 52.16 | 48.21 | 27.99 | 74.83 | 21.33 | 22.72 | 43.41 | 24.06 | 18.24 | 11.93 | 3.51 | 63.99 | 48.68 | 14.68 | 47.92 | 34.91 |

Table 3: Zero-shot clean accuracy under PGD-100 with $\epsilon = 1/255$ on 15 datasets. All methods are fine-tuned on TinyImageNet using PGD-2.

| Methods | Tiny-ImageNet | CIFAR-10 | CIFAR-100 | STL-10 | SUN397 | Food101 | OxfordPets | Flowers102 | DTD | EuroSAT | FGVCAircraft | Caltech-101 | Caltech-256 | StanfordCars | PCAM | Average |
|---|---|---|---|---|---|---|---|---|---|---|---|---|---|---|---|---|
| CLIP | 57.26 | 88.06 | 60.45 | 97.04 | 57.26 | 83.89 | 87.41 | 65.47 | 40.69 | 42.59 | 20.25 | 85.34 | 81.73 | 52.02 | 52.09 | 64.77 |
| FT-Clean | 79.04 | 84.55 | 54.25 | 93.78 | 46.80 | 80.98 | 46.33 | 30.32 | 24.39 | 9.30 | 9.30 | 78.69 | 70.81 | 31.15 | 47.89 | 52.51 |
| FT-Adv | 73.83 | 68.96 | 39.69 | 86.89 | 33.37 | 27.74 | 60.10 | 33.45 | 13.26 | 16.49 | 4.86 | 67.41 | 57.72 | 18.11 | 49.91 | 43.45 |
| TeCoA | 63.97 | 66.14 | 36.74 | 87.24 | 40.54 | 35.11 | 66.15 | 33.25 | 13.75 | 17.13 | 6.75 | 64.63 | 56.20 | 25.65 | 49.01 | 44.15 |
| FARE | 77.54 | 87.58 | 62.80 | 94.33 | 49.91 | 70.02 | 81.47 | 57.10 | 36.33 | 22.69 | 14.19 | 84.04 | 77.50 | 44.35 | 46.07 | 60.39 |
| PMG-AFT | 67.11 | 74.62 | 44.68 | 88.85 | 37.42 | 37.47 | 66.34 | 35.66 | 21.17 | 17.76 | 4.71 | 76.70 | 61.96 | 25.21 | 49.60 | 47.28 |
| CAW | 75.64 | 82.96 | 55.49 | 91.36 | 41.96 | 50.87 | 71.02 | 42.15 | 28.56 | 23.42 | 9.42 | 80.66 | 67.94 | 34.88 | 49.98 | 53.75 |

# C    Ablation studies

To evaluate our method, we compare against the reported results of CLIP, FT-Clean, FT-Adv, TeCoA, FARE [41], PMG-AFT, and TGA-ZSR, as presented in the TGA-ZSR paper [19]. FT-Clean and FT-Adv are fine-tuned using clean and adversarial examples, both with contrastive loss.

**PGD and CW Attack**    As shown in Table 2, our method outperforms PMG-AFT in robust accuracy on most datasets, achieving a higher average performance. Table 3 further demonstrates that our method surpasses PMG-AFT in clean accuracy across all datasets. Based on these results, our method performs well on both clean and adversarial samples, showing competitive performance compared to other approaches. Table 4 indicates that our approach achieves better results than PMG-AFT under the CW attack. We compare only with CLIP and PMG-AFT because these are the only methods reported in the paper [19]. The model is trained with PGD-2 using a perturbation bound of $\epsilon = 1/255$ and evaluated on PGD-100 and CW with the same bound.

**Effect of Attack Strength**    Table 5 presents the average robust accuracy under PGD-100 with perturbation bounds of $\epsilon = 1/255$, $2/255$, and $4/255$ across 15 datasets. Our method outperforms PMG-AFT on average and surpasses other baseline methods across various attack strengths.

**Analyzing the Effect of Each Loss Component**    As shown in Table 6, the $L_{CE}$ row reports the average clean and robust accuracy across all 15 datasets under PGD-100 with $\epsilon = 1/255$. The $L_{CA}$ row presents the results after adding this component to the previous loss term. Finally, the $L_{Reg}$ row reflects the performance using the full loss function. These results demonstrate that our method improves both robustness and clean accuracy on average, compared to the standard CLIP loss.

**Analysis of Computational Cost and Memory Usage**    As shown in Table 7, our method uses less memory than both PMG-AFT and TGA-ZSR while achieving better accuracy under stronger attacks, as discussed in previous sections. It also maintains a training time comparable to the aforementioned approaches.

Table 4: Zero-shot robust accuracy under CW attack on 15 datasets. All methods are fine-tuned on TinyImageNet using PGD-2.

| Methods | Tiny-ImageNet | CIFAR-10 | CIFAR-100 | STL-10 | SUN397 | Food101 | OxfordPets | Flowers102 | DTD | EuroSAT | FGVCAircraft | Caltech-101 | Caltech-256 | StanfordCars | PCAM | Average |
|---|---|---|---|---|---|---|---|---|---|---|---|---|---|---|---|---|
| CLIP | 0.21 | 0.36 | 0.10 | 10.59 | 1.16 | 0.82 | 1.23 | 1.09 | 2.18 | 0.01 | 0.00 | 13.50 | 7.36 | 2.36 | 0.07 | 2.45 |
| PMG-AFT | 44.59 | 44.86 | 24.15 | 74.11 | 19.99 | 17.33 | 39.88 | 20.95 | 13.51 | **12.09** | 1.47 | 60.99 | 44.46 | 10.57 | **48.59** | 32.36 |
| CAW | **51.7** | **47.68** | **26.80** | **74.62** | **20.46** | **21.52** | **43.79** | **22.29** | **16.22** | 11.60 | **3.51** | **63.48** | **47.91** | **14.09** | 47.71 | **34.87** |

Table 5: Zero-shot robust accuracy under PGD-100 with $\epsilon = 1/255, 2/255$ and $4/255$ on 15 datasets. All methods are fine-tuned on TinyImageNet using PGD-2.

| Methods | Tiny-ImageNet | CIFAR-10 | CIFAR-100 | STL-10 | SUN397 | Food101 | OxfordPets | Flowers102 | DTD | EuroSAT | FGVCAircraft | Caltech-101 | Caltech-256 | StanfordCars | PCAM | Average |
|---|---|---|---|---|---|---|---|---|---|---|---|---|---|---|---|---|
| CLIP | 0.64 | 2.15 | 0.12 | 20.35 | 0.52 | 5.94 | 2.97 | 0.72 | 0.71 | 0.03 | 0.00 | 14.28 | 9.18 | 0.11 | 0.04 | 3.65 |
| FT-Clean | 12.44 | 18.80 | 4.65 | 37.16 | 0.43 | 0.52 | 2.03 | 0.41 | 0.92 | 0.02 | 0.01 | 13.02 | 7.96 | 0.03 | 0.44 | 6.21 |
| FT-Adv | 29.33 | 18.10 | 11.06 | 45.13 | 8.58 | 5.65 | 16.45 | 10.15 | 9.72 | 9.82 | 0.83 | 33.43 | 24.14 | 3.80 | 38.06 | 17.07 |
| TeCoA | 18.17 | 12.78 | 8.12 | 39.87 | 8.53 | 6.12 | 11.04 | 10.07 | 10.07 | 9.88 | 0.63 | 34.94 | 23.92 | 3.45 | 33.20 | 15.41 |
| FARE | 12.41 | 9.09 | 4.23 | 33.72 | 2.98 | 4.75 | 9.67 | 5.52 | 4.26 | 0.25 | 0.28 | 23.97 | 16.95 | 1.48 | 3.43 | 8.54 |
| PMG-AFT | 25.30 | 21.71 | 13.29 | 47.69 | **11.42** | 9.49 | **20.68** | **12.86** | 9.45 | **10.65** | 0.90 | **41.86** | 28.92 | 3.72 | **37.88** | 19.27 |
| CAW | **31.15** | **22.49** | **13.67** | **47.99** | 9.87 | **9.88** | 20.16 | 12.14 | **10.90** | 7.05 | **1.43** | 41.33 | **29.06** | **6.03** | 29.88 | **19.53** |

# D Discussion

**Limitations**   Our method focuses solely on the CLIP model and has not been tested on other vision-language models under adversarial attacks. In addition, it only addresses adversarial perturbations in the image encoder, whereas the text encoder is also a crucial component of VLMs and should be considered to improve overall robustness.

**Broder impact**   Large-scale models like VLMs have demonstrated strong zero-shot capabilities, performing well across diverse tasks and unseen domains. However, their performance under adversarial perturbations remains limited, which is an important and active area of research. As these models are increasingly deployed in real-world applications, ensuring their robustness and privacy against adversarial attacks becomes critical. Our method aims to improve the zero-shot robustness of CLIP under such attacks, contributing to the development of safer and more reliable vision-language systems.

Table 6: Average zero-shot robust and clean accuracy after adding each component, evaluated under PGD-100 with $\epsilon = 1/255$ on 15 datasets. All methods are fine-tuned on TinyImageNet using PGD-2.

| | Robust | Clean | Average |
|---|---|---|---|
| CLIP | 4.90 | 64.42 | 34.66 |
| $\mathcal{L}_{CE}$ | 30.39 | 45.58 | 37.98 |
| $+\mathcal{L}_{CA}$ | 33.64 | 51.50 | 42.57 |
| $+\mathcal{L}_{Reg}$ | 34.92 | 53.65 | 44.28 |

Table 7: Memory Consumption and Training Time

| Methods | Train memory usage | Train time (per epoch / batch) |
|---|---|---|
| CLIP | 0Mb | 0s / 0s |
| TeCoA | 12,873Mb | 512s / 0.65s |
| CAW | 15,986Mb | 842s / 1.08s |
| PMG-AFT | 18,449Mb | 828s / 1.06s |
| TGA-ZSR | 21,227Mb | 885s / 1.13s |

