# OpenReview forum: "Zero-Shot Robustness of Vision Language Models Via Confidence-Aware Weighting"
_NeurIPS.cc/2025/Workshop/Reliable_ML — NeurIPS 2025 - Reliable ML Workshop_

### Official Review · Reviewer_oy1C · 2025-09-15
**Strong Empirical Results, Missing Theory**

**Rating:** 7
**Confidence:** 3

**Review:**

### **Summary**
The paper addresses the vulnerability of CLIP to adversarial attacks in zero-shot settings. The authors propose Confidence-Aware Weighting, which consists of (1) a confidence-aware loss that emphasizes adversarial samples where the model is less confident, and (2) a regularization that preserves semantic knowledge by aligning adversarial features of frozen and fine-tuned encoders. Experiments across 15 datasets show that CAW achieves state-of-the-art robustness under AutoAttack, PGD-100, and CW, while maintaining or improving clean accuracy and using less memory than existing baselines.

### **Strength**
* Novelty: The confidence-aware weighting formulation is a conceptual contribution, extending ideas from ARoW but tailored to VLMs.
* Empirical Quality: Extensive experiments on 15 datasets, strong baselines, and ablation studies (showing contributions of each loss component) strengthen the empirical evidence.
* Clarity: The methodology is clearly described with equations, diagrams. The motivation is clear
* Relevance: Highly relevant to the community’s interest in adversarial robustness.

### **Weaknesses/Limitations**
* KL divergence direction: The method departs from prior literature by using $\mathrm{KL}(P_{\text{adv}} || P_{\text{clean}})$ instead of $\mathrm{KL}(P_{\text{clean}} || P_{\text{adv}})$. However, this adaptation is only empirically motivated — the paper does not provide a theoretical explanation or analysis of why the reversal is preferable.
* Hyperparameter sensitivity: Add experiments showing how α and β affect results, or justify their values with more systematic tuning.
* Narrow scope: Experiments are limited to CLIP (ViT-B/32 backbone). No evidence of applicability to other architectures (e.g., BLIP, ALIGN). (mentioned by authors)
* Text encoder untouched: Method only addresses adversarial robustness in the image encoder, while ignoring vulnerabilities in the text encoder. (mentioned by authors)
* Interpretability: While robustness improves, the paper offers limited insights into why the method works beyond performance metrics.

### **Suggestions for Authors**
* KL divergence justification: The choice of KL divergence direction is theoretically motivated in the ARoW paper. It would be ideal if we have theoretical motivation for the flip here.
* Hyperparameter sensitivity: Add experiments showing how α and β affect results, or justify their values with more systematic tuning.
* Generalization beyond CLIP: Apply CAW to other large VLMs (e.g., BLIP, ALIGN) to validate generality.
* Adversarial text perturbations: Extend robustness to the text encoder; many real-world attacks exploit multimodal vulnerabilities.
* Efficiency trade-offs: Training time is slightly higher than PMG-AFT. A deeper discussion of compute-accuracy trade-offs would be valuable.

---

### Official Review · Reviewer_af6J · 2025-09-20
**Confidence-Aware Fine-Tuning Improves Zero-Shot Robustness of CLIP**

**Rating:** 6
**Confidence:** 4

**Review:**

**Summary**
The paper proposes *Confidence-Aware Weighting (CAW)*, an adversarial fine-tuning method for CLIP that emphasizes uncertain adversarial examples and aligns features with a frozen encoder. Experiments on 15 datasets show CAW improves both clean and robust accuracy under strong attacks (AutoAttack, PGD-100, CW) while reducing memory use compared to prior methods.

**Strengths**
* Addresses a highly relevant reliability problem: the extreme vulnerability of zero-shot VLMs to adversarial perturbations.
* The proposed method is conceptually simple yet effective, requiring only fine-tuning with a modified loss.
* Empirical evaluation is extensive: 15 datasets, multiple attack settings (AutoAttack, PGD, CW), and comparisons with strong baselines.
* Results demonstrate improvements in both clean and robust accuracy.

**Weaknesses**
* Evaluation focuses solely on the CLIP ViT-B/32 backbone and image encoder; no evidence that the approach generalizes to other VLM architectures or text-side vulnerabilities.
* The paper does not analyze **failure cases**: when does CAW break down, e.g., with stronger perturbation budgets, transfer attacks, or adaptive adversaries?

**Suggestions for Authors**
Please resolve the weaknesses.